# Oral Administration of Glutathione Trisulfide Increases Reactive Sulfur Levels in Dorsal Root Ganglion and Ameliorates Paclitaxel-Induced Peripheral Neuropathy in Mice

**DOI:** 10.3390/antiox11112122

**Published:** 2022-10-27

**Authors:** Mariko Ezaka, Eizo Marutani, Yusuke Miyazaki, Eiki Kanemaru, Martin K. Selig, Sophie L. Boerboom, Katrina F. Ostrom, Anat Stemmer-Rachamimov, Donald B. Bloch, Gary J. Brenner, Etsuo Ohshima, Fumito Ichinose

**Affiliations:** 1Anesthesia Center for Critical Care Research, Department of Anesthesia, Critical Care and Pain Medicine, Massachusetts General Hospital, Boston, MA 02114, USA; 2Department of Pathology, Massachusetts General Hospital, Boston, MA 02114, USA; 3Department of Medicine, Division of Rheumatology, Allergy and Immunology, Massachusetts General Hospital, Boston, MA 02114, USA; 4Corporate Strategy Department, Kyowa Hakko Bio Co., Ltd., Tokyo 164-0001, Japan

**Keywords:** paclitaxel-induced peripheral neuropathy, glutathione trisulfide, persulfide, persulfidation

## Abstract

Peripheral neuropathy is a dose-limiting side effect of chemotherapy with paclitaxel. Paclitaxel-induced peripheral neuropathy (PIPN) is typically characterized by a predominantly sensory neuropathy presenting with allodynia, hyperalgesia and spontaneous pain. Oxidative mitochondrial damage in peripheral sensory neurons is implicated in the pathogenesis of PIPN. Reactive sulfur species, including persulfides (RSSH) and polysulfides (RS_n_H), are strong nucleophilic and electrophilic compounds that exert antioxidant effects and protect mitochondria. Here, we examined the potential neuroprotective effects of glutathione trisulfide (GSSSG) in a mouse model of PIPN. Intraperitoneal administration of paclitaxel at 4 mg/kg/day for 4 days induced mechanical allodynia and thermal hyperalgesia in mice. Oral administration of GSSSG at 50 mg/kg/day for 28 days ameliorated mechanical allodynia, but not thermal hyperalgesia. Two hours after oral administration, ^34^S-labeled GSSSG was detected in lumber dorsal root ganglia (DRG) and in the lumber spinal cord. In mice treated with paclitaxel, GSSSG upregulated expression of genes encoding antioxidant proteins in lumber DRG, prevented loss of unmyelinated axons and inhibited degeneration of mitochondria in the sciatic nerve. In cultured primary neurons from cortex and DRG, GSSSG mitigated paclitaxel-induced superoxide production, loss of axonal mitochondria, and axonal degeneration. These results indicate that oral administration of GSSSG mitigates PIPN by preventing axonal degeneration and mitochondria damage in peripheral sensory nerves. The findings suggest that administration of GSSSG may be an approach to the treatment or prevention of PIPN and other peripheral neuropathies.

## 1. Introduction

In 2018, approximately 9.8 million cancer patients were treated with chemotherapy; the number of patients requiring cancer chemotherapy is expected to reach 15 million/year in 2040 [1]. Paclitaxel (PTX) is one of the most commonly used chemotherapeutic agents and is part of regimens to treat breast, ovarian, and prostate cancers. Paclitaxel-induced peripheral neuropathy (PIPN) is a dose-limiting side effect of paclitaxel, affecting 30% to 70% of treated patients [2]. PIPN manifests as allodynia, hyperalgesia and spontaneous pain, predominantly involving feet and hands [3]; PIPN generally develops during chemotherapy and often persists after cessation of paclitaxel [4]. PIPN worsens the quality of life of cancer survivors and can be severe enough to require discontinuation of chemotherapy. Unfortunately, other chemotherapeutic agents, including cisplatin and vincristine, can also cause peripheral neuropathy (termed “chemotherapy-induced peripheral neuropathy” (CIPN)). These chemotherapies may produce symptoms that are similar to those seen in patients with PIPN and are thought to act via similar pathogenetic mechanisms [5]. Because of the growing number of cancer survivors worldwide [1], the medico-economic impact of CIPN has increased significantly. Currently, there are few treatment options available for treating CIPNs, including PIPN.

While the precise pathogenesis of PIPN is unknown, the disease is thought to target peripheral sensory neuronal axons [6,7,8], which would explain why the longer axons of the hands and feet are predominantly affected. Impairment of Aβ fibers leads to mechanical allodynia, while hyperalgesia arises from damaged myelinated Aδ and unmyelinated C fibers [9]. The mechanism of action of paclitaxel as a chemotherapeutic agent is inhibition of cancer cell proliferation by stabilizing microtubule polymers, which need to disassemble during mitosis. In addition to this anti-tumor effect, paclitaxel causes mitochondrial dysfunction, which manifests as mitochondrial swelling [10], decreased mitochondrial membrane potential [11,12], increased levels of reactive oxygen species (ROS) [13], and impaired oxidative phosphorylation in peripheral neurons [14]. These off-target effects of paclitaxel have been implicated in the pathogenesis of PIPN. Nonetheless, no therapies have been developed to date that protect mitochondria in peripheral nerve sensory axons.

Reactive sulfur species, including persulfides and polysulfides, contain reactive sulfur that oxidizes or reduces other molecules. In particular, sulfane sulfur (S^0^) atom has strong nucleophilicity that promotes persulfidation of protein thiols (cysteine residue). Thiol persulfidation competes with ROS-mediated thiol oxidation, thereby protecting proteins from irreversible oxidation. Endogenous persulfides, such as glutathione persulfide (GSSH) and cysteine persulfide (CysSSH), exert potent antioxidant effects, playing a key role in maintaining intracellular redox balance [15]. Nonetheless, the possible role of systemic administration of polysulfides to protect peripheral sensory neurons from paclitaxel injury by increasing local concentrations of reactive sulfur has not yet been examined. This study was designed to address this knowledge gap by examining the effects of a systemically administered stable formulation of glutathione trisulfide (GSSSG), an endogenous polysulfide, in a mouse model of PIPN. The structure of GSSSG contains a sulfane sulfur (Figure 1, arrow), and GSSSG is in equilibrium with GSSH as shown in the following equation [16].
GSSSG + GSH ↔ GSSH + GSSG(1)

We hypothesized that systemic administration of GSSSG would increase the local concentration of reactive sulfur species in peripheral sensory nerves and ameliorate PIPN by protecting sensory nerve mitochondria. This study also examined whether GSSSG can attenuate PIPN without reducing the anti-tumor effects of paclitaxel.

## 2. Materials and Methods

### 2.1. Animals

All animal protocols were approved by the Massachusetts General Hospital Institutional Animal Care and Use Committee (protocol No. 2020N000126). Animals were cared for in accordance with the guidelines established by the NIH and the International Association for the Study of Pain [17]. Male C57BL/6J mice (6–7 weeks old) were purchased from the Jackson Laboratory (Bar Harbor, ME, USA). The mice were housed in a 12-h shift light-control environment from 7 a.m. to 7 p.m. with ad libitum access to food and water in our animal facility until the time of experiments.

### 2.2. Drugs and Animal Models

Paclitaxel (Sigma-Aldrich, St. Louis, MO, USA) was dissolved with ethanol and cremophor (1:1) and diluted with normal saline (1:4). Peripheral neuropathy was induced in mice by intraperitoneal (i.p.) administration of 4 mg/kg paclitaxel every other day for a total of 4 injections (cumulative dose of 16 mg/kg), according to a previously described protocol [18]. A stable, crystallized form of GSSSG was produced and provided by Kyowa Hakko Bio CO., Ltd. (Tokyo, Japan). GSSSG was pulverized and mixed in 0.5% methylcellulose. To evaluate the therapeutic effects of GSSSG in a model of PIPN, mice were randomly divided into three groups of 6. Mice were treated with: (1) paclitaxel alone; (2) paclitaxel and GSSSG; (3) or vehicle alone (control). Mice in groups 1 and 2 received paclitaxel as described above, while mice in the control group received the same volume of vehicle. The first dose of GSSSG was given within 1 h after the first PTX injection. Treatment was administered as oral gavage for 28 days with GSSSG at 50 mg/kg/day (group 2) or 0.5% methylcellulose (groups 1 and 3). The dose of GSSSG was determined based on pilot studies. To determine the effect of GSSSG on the development of PIPN, behavioral tests to assess allodynia and hyperalgesia were conducted on all mice on days −1 (baseline), 7, 14, 21, and 28 after paclitaxel administration. Behavioral tests were performed by an investigator who was blinded to the treatment groups, as described in the next section.

To determine the tissue distribution of orally administered GSSSG, ^34^S-labelled GSSSG was administered by gavage to each of four mice. ^34^S-labelled GSSSG was synthesized and provided by Kyowa Hakko Bio Co., Ltd. (Tokyo, Japan). Two hours after a single dose of ^34^S-GSSSG, DRG, lumber spinal cord, brain, liver, and plasma were obtained and snap frozen for later analysis.

### 2.3. Behavior Testing

#### 2.3.1. Measurement of Mechanical Allodynia—Von Frey Filament Test

To assess allodynia, mechanical withdrawal thresholds were measured by the manual von Frey filament test. Before testing, mice were habituated for 3–4 days and acclimated in the test apparatus (a plastic cage on a wire mesh floor) for 30 min for 3 consecutive days. An investigator who was blinded to treatment group stimulated the mid-plantar surface of their hind paws with von Frey filaments (Ugo Basile, Gemonio, Italy) in the range from 0.04 g force (g) to 2.0 g. Filaments were applied with constant speed until they bent. Behavioral responses, withdrawal and licking of their paws, were considered reactive. The stimulation of the same size filament was repeated up to 2 times if the mouse did not react. Mice were initially tested with a 0.6 g filament and subsequent filament size was determined based the response. A smaller filament was used after a positive response and a larger filament was used after a negative response. Filaments were applied up to a total of 9 times. The 50% threshold of a paw withdrawal response was calculated using the Up-down Reader (an open-source program, ver. 2.0) [19] based on the up and down method [20]. This behavior test was conducted weekly for 4 weeks.

#### 2.3.2. Measurement of Thermal Hyperalgesia—Hot Plate Test

To assess thermal hyperalgesia, the heat threshold was determined by the hot plate test [21]. Mice were placed on the hot plate (Hot/Cold Plate NG, Ugo Basile, Gemonio, Italy) warmed to 52 °C. Response latency was the time required to exhibit nociceptive behavior, including hind paw withdrawal or licking, stamping, and jumping. The test was repeated 2 times after 5 min interval by a blinded examiner, and the average time was calculated. For each test, the ratio of change from baseline was determined, based on the difference in the latency response time from the baseline divided by the baseline time. This test was conducted weekly for 4 weeks.

### 2.4. Immunohistochemistry Staining of Intraepidermal Nerve Fibers

The density of intraepidermal nerve fibers (fibers/mm) in the hind paws was calculated to evaluate peripheral nerve fiber damage by paclitaxel. Mice were deeply anesthetized with Isoflurane (4%) and euthanized by exsanguination on week 1 or week 4 after paclitaxel treatment. Mice were perfused with cold 4% paraformaldehyde in PBS through the left ventricle. Skin from the hind paws was harvested and fixed in 4% paraformaldehyde overnight, then cryoprotected in 20% sucrose solution at 4 °C until they sank and then in 30% sucrose solution overnight at 4 °C. Tissue blocks were then submerged in optimal cutting temperature medium, frozen at −80 °C, and sectioned using a cryostat (25 μm thickness). The sections were then incubated with 0.3% hydrogen peroxide for 10 min and with blocking solution (5% donkey serum; Sigma-Aldrich, St. Louis, MO, USA, 0.3% Triton-114) for one-hour. Sections were then incubated overnight at 4 °C with anti-protein gene product (PGP) 9.5 antibody (1:100, rabbit, Abcam, Cambridge, UK) and anti-collagen IV antibody (1:400, goat, SouthernBiotech, AL, USA). Sections were subsequently incubated with secondary antibodies (1:300, donkey anti-rabbit, Abcam, Cambridge, UK; donkey anti-goat, Abcam) for one-hour at room temperature and covered with fluoroshield mounting medium with DAPI (Sigma-Aldrich, St. Louis, MO, USA). Ten images per mouse (6 mice per group) were taken by fluorescence microscopy (Nikon Eclipse 80i, Nikon Instruments, Inc., Melville, NY, USA), and four of the ten images were randomly selected. The density (fibers/mm basement membrane) was calculated as the number of intraepidermal nerve fiber divided by the length of basement membrane, in accordance with published guidelines [22].

### 2.5. Histological Evaluation of Sciatic Nerves

#### 2.5.1. Toluidine Blue Staining

Four weeks after paclitaxel treatment, sciatic nerves were evaluated by toluidine blue staining as described previously [23]. Briefly, mice were deeply anesthetized and placed in a prone position to facilitate exposure of the sciatic nerves. Sciatic nerves were covered with Trump’s fixative (Quimigen, Madrid, Spain) for 10 min. Subsequently, nerves were harvested and fixed in the same fixative for one week changing fixative every other day. Samples were immersed in 2% osmium tetroxide (TGI, Dallas, TX, USA) for 2 h and embedded in Resin-Epoxy medium (Sigma-Aldrich, St. Louis, MO, USA) overnight at 60 °C by following the protocol provided by the manufacturer. The embedded nerve block was sectioned at 1 μm by an ultramicrotome (Reichert-Jung, Ultracut E, Vienna, Austria) and stained in 1% toluidine blue (Sigma-Aldrich, St. Louis, MO, USA). Two images per mouse (6 mice per group) were taken by light microscopy (Nikon Eclipse 80i, Nikon Instruments, Inc., Melville, NY, USA) and analyzed by ImageJ. Thickness of myelin was evaluated using the G-ratio, which compares the size of the inner radius and outer radius. G-ratio was calculated using GRatio for ImageJ, an ImageJ (ver. 1.53, NIH, Bethesda, MD, USA) plugin available online (http://gratio.efil.de/ (accessed on 1 April 2021)) that converts inner and outer perimeter of myelin to the radius. Counting the number of axons and measuring the size of each axon was performed by an examiner who was blinded to the treatment of mice.

#### 2.5.2. Transmission Electron Microscopy

To evaluate unmyelinated axons and mitochondria, we examined sciatic nerve axons by transmission electron microscopy. Resin-embedded nerve blocks were sectioned at 50 nm using an ultramicrotome. Sections were examined with an FEI Morgagni transmission electron microscope (FEI, Lausanne, Switzerland). Low (×2200) and high (×11,000) magnification of images were captured with an AMT 2K charge-coupled device camera (Advanced Microscopy Techniques, Woburn, MA, USA). Unmyelinated axons were quantified using 9 low magnified images per group (3 mice per group). The percentage of unmyelinated axons was calculated by dividing unmyelinated axons by the total number of axons. The cross-sectional area of mitochondria (μm^2^) in unmyelinated axons was quantified in 18 to 20 high magnified images per group (3 mice per group). In total, 80 to 100 mitochondrion per group were evaluated. Quantification was conducted by a blinded examiner, and the images were evaluated by a pathologist (A.S.-R.) who was also blinded to the identity of samples.

### 2.6. Real-Time Quantitative Polymerase Chain Reaction (qPCR)

Paclitaxel alone- and paclitaxel and GSSSG-induced changes in gene expression in lumber dorsal root ganglions (DRG) were examined using real-time qPCR after a single administration of paclitaxel with or without GSSSG. We simultaneously administered 16 mg/kg paclitaxel by i.p. injection and 50 mg/kg GSSSG by oral gavage and determined the level of mRNA 2 h later. Lumbar DRG were isolated from mice as described previously [24]. DRG were immersed in “RNA later” (Invitrogen, Waltham, MA, USA) at 4 °C overnight. After retrieval from RNA later, samples were stored at −80 °C. DRGs were homogenized in TRIzol reagent (Thermo Scientific Scientific, Waltham, MA, USA) and mixed with chloroform. After centrifugation at 14,000× *g* at 4 °C for 15 min, the transparent top layer was transferred to a new tube. The samples were mixed with 400 μL of isopropanol and incubated at −20 °C for 20 min. After centrifugation at 14,000× *g* at 4 °C for 15 min, the pellets were collected, mixed with 70% ethanol, and centrifuged at 14,000× *g* at 4 °C for 10 min. The pellets were dried for 10 min and incubated with 50 μL of nuclease-free water. Complementary DNA was synthesized using the cDNA Reverse Transcription Kit (Applied Biosystems, Waltham, MA, USA) and quantitative PCR was performed using SYBR green (Applied Biosystems, Waltham, MA, USA). Primers are listed in Appendix A Table A1. Relative quantification of gene expression was performed via 2^−ΔΔCT^ method.

### 2.7. Mass Spectrometry to Detect GSSSG Administered by Oral Gavage

We used liquid chromatography with tandem mass spectrometry (LC-MS/MS) to determine whether GSSSG, administered via oral gavage, reaches the peripheral nervous system. Fifty mg/kg of ^34^S-labeled GSSSG (in which the middle sulfur, that is a sulfane sulfur, was replaced to ^34^S; G-^32^S-^34^S-^32^S-G) was orally administered to mice. Two hours after administration, plasma and tissue from liver, brain, lumbar spinal cord, and lumbar DRG (L1 to L6) were collected and immediately frozen at −80 °C. Tissues were homogenized with 5 mM β-(4-hydroxyphenyl)ethyl iodoacetamide (HPE-IAM) (Santa Cruz, Dallas, TX, USA) and incubated for 20 min at 37 °C to promote the HPE-IAM reaction, which stabilizes persulfide residues [25]. Proteins were removed by centrifugation at 15,000×*g* for 10 min at 4 °C, and total protein concentration was measured by BCA assay. The supernatant was diluted by 0.1% formic acid for LC-MS/MS analysis. The amount of ^34^S-labeled GSSSG was quantified in selective reaction monitoring (SRM) with precursor ion (647.14 *m*/*z*), product ion (389.1 *m*/*z*), and HCD (21 v) and normalized by protein concentration. The ratio of ^34^S-labeled reactive sulfur species to endogenous (^32^S) reactive sulfur species (GSSH, CysSSH, and CysSSSCys) was calculated from their peak areas, measured by Dionex UltiMate 3000 RS UPLC-Orbitrap Exploris 480 mass spectrometer (Thermo Scientific Scientific, Waltham, MA, USA). In brief, samples were subjected to the UPLC system with a Hypersil Gold C-18 (100 × 2.1 mm, 3.0 μm, Thermo Fisher Scientific) column and components were eluted using a linear methanol gradient of the mobile phase (0–90%, 15 min) in the presence of 0.1% formic acid at a flow rate of 0.2 mL/min at 40 degrees. The raw data were obtained by Compound Discoverer software 3.3 (Thermo Scientific Scientific, Waltham, MA, USA). The molecular weight of reactive sulfur species combined with HPE-IAM were reported in previous studies [26,27].

### 2.8. In Vitro Studies

#### 2.8.1. Primary DRG Neuron Isolation and Histological Evaluation

Primary DRG neurons were prepared from 8 to 10-week old mice as described previously [28]. Briefly, lumber DRG (L1 to L6) were isolated from mice and incubated with Dispase-II Solution (Sigma-Aldrich, St. Louis, MO, USA) and collagenase type II (Worthington, Columbus, OH, USA) for 70 min and with 0.25% trypsin for 5 min. DRG were triturated using a frame polished glass pipet, and neurons were seeded on a 12-well plate with 15 mm coverslip. Cells were incubated in Neurobasal A-medium with 2% B27 supplement (Gibco, Waltham, MA, USA), 1% penicillin/streptomycin, 1% Glutamax (Gibco, Waltham, MA, USA), and nerve growth factor (Sigma Aldrich, St. Louis, MO, USA) for 24 h. Cells were then exposed to 100 nM of paclitaxel, with or without GSSSG (500 nM), for one-hour and with MitoTracker (50 nM) (Invitrogen, Waltham, MA, USA) for 30 min. After fixing in 4% paraformaldehyde for 10 min at room temperature, cells were immersed in 0.2% Triton X-100 in PBS for 7 min and washed with PBS. Cells were then incubated with anti-NF200 antibody (1:400, mouse, Sigma-Aldrich, St. Louis, MO, USA) overnight at 4 °C and with the secondary antibody (1:1000, donkey anti-mouse, Abcam, Cambridge, UK) for one-hour at room temperature. After being covered with fluoroshield mounting medium with DAPI (Sigma-Aldrich, St. Louis, MO, USA), 9–10 cells per group (3 mice per group) were imaged using confocal microscopy (ZEISS LSM 800, Carl Zeiss, Thornwood, NY, USA) with a 63× oil immersion objectives lens with 1024 × 1024 pixels image size.

#### 2.8.2. Primary Cortical Neuron Isolation and ROS Assay

Primary cortical neurons were prepared from mice on embryonic day 15, as described previously [29]. In brief, each embryo’s cortex was isolated in Hanks’ balanced salt solution and centrifuged at 176× *g* (1000 rpm) for 3 min. After aspirating the solution, cells were incubated with 0.25% trypsin for 15 min and cells were seeded at a density of 20,000 cells/well in a 96-well plate coated with the poly-D-lysine (Gibco, Waltham, MA, USA). Cells were incubated in Neurobasal medium (Gibco, Waltham, MA, USA) with 2% B27 supplement (Gibco, Waltham, MA, USA), 1% penicillin/streptomycin, and 1% Glutamax (Gibco, Waltham, MA, USA) until day 11, when they were used in experiments.

To examine whether GSSSG reduces ROS generated by paclitaxel, we conducted a dihydroethidium (DHE) assay (Abcam, Cambridge, UK) that is sensitive to superoxide. The experiment was performed according to the protocol provided by the manufacturer. Briefly, primary cortical neurons were incubated in DHE reagent with or without 10 and 30 μM of GSSSG for 30 min. Cells were subsequent treated with paclitaxel (100 nM) for one-hour. Fluorescence of DHE was measured at 490 nm of excitation and 585 nm of emission wavelength.

### 2.9. Cancer Cell Line and Viability Assay

To examine whether administration of GSSSG affects anti-tumor effects of paclitaxel, we conducted experiments using the human breast cancer cell line MDA-MB-231 (HTB-26, ATCC). Cells were cultured in medium consisting of 90% RPMI1640 (Corning, Corning, NY, USA), 10% FBS, and 1% penicillin/streptomycin. After being seeded into a 96-well plate at the density of 20,000 cells/well and cultured overnight, cells were exposed to different concentrations of paclitaxel (0.125 μM, 0.25 μM, 0.5 μM, 1 μM, and 2 μM) and incubated for 24 h. Cell viability was evaluated using the LDH Cytotoxicity Detection Kit (Roche, Basel, Switzerland). Cells were washed with PBS and incubated with 100 μL of 1% Triton X-100 at 37 °C for 30 min. After mixing with assay enzyme for 30 min at 25 °C, absorbance was measured at wavelength 492 nm to determine cell viability. The half-maximal inhibitory concentration (IC_50_) of paclitaxel was determined. Using the IC_50_ of paclitaxel, we determined the effect of GSSSG on the anti-tumor effect of paclitaxel using the trypan blue exclusion assay. MDA-MB-231 cells were seeded at 5.6 × 10,000 cells/well in a 6-well plate and incubated overnight. The cells were treated with paclitaxel (at the IC_50_ concentration), with or without GSSSG (10 μM) for 24 h. After suspension using 0.25% trypsin, cells were treated with 0.04% trypan blue and the percentage of cells that retained the ability to exclude blue dye was determined. Five wells were evaluated per group.

### 2.10. Statistical Analysis

Sample sizes for behavior tests were chosen based on a previous study [30]. All values are expressed as mean ± standard deviation (SD). Behavior test results were analyzed in the mixed effect model because the behavior was measured repeatedly five times in each animal, and individual differences at baseline were confirmed in the preliminary study. Bonferroni correction was applied to correct for multiple comparisons in the mixed effect model. Parametric data were analyzed by one-way analysis of variance (ANOVA) with Dunnett’s multiple comparisons test. Non-parametric data were analyzed by Kruskal–Wallis test with Dunn’s multiple comparisons test. DRG morphology was analyzed by two-way repeated measures ANOVA with Dunnett’s multiple comparisons test. Anti-tumor effect of paclitaxel with GSSSG was analyzed by equivalence test using two one-sided *t*-tests. The margin of equivalence was defined as 10% difference in cell count. Probability (*p*) value less than 0.05 was considered significant. Statistical analyses were performed using GraphPad Prism 9.1 (GraphPad Software Inc., La Jolla, CA, USA).

## 3. Results

### 3.1. GSSSG Prevented Paclitaxel-Induced Mechanical Allodynia

Male adult mice were treated with 4 mg/kg paclitaxel i.p. on days 0, 2, 4, and 6. Starting on day 7 and continuing through week 4, paclitaxel-treated mice demonstrated mechanical allodynia as assessed using von Frey testing of the hind paw (Figure 2A closed circles, *p* = 0.0014). Paclitaxel also induced thermal hyperalgesia as measured using the hot plate test (Figure 2B, closed circles, *p* < 0.0001). Oral administration of GSSSG at 50 mg/kg/day mitigated mechanical allodynia over the experimental period (Figure 2A, black squares, *p* = 0.003). In contrast, thermal hyperalgesia was not altered by GSSSG administration (Figure 2B, black squares).

### 3.2. GSSSG Prevented Loss of Intraepidermal Nerve Fibers Induced by Paclitaxel

We evaluated degeneration of unmyelinated nerve endings by determining the density of intraepidermal nerve fibers, which is a widely used indicator of peripheral neuropathy. The density of the intraepidermal nerve fibers in the planter surface skin of hind paws of mice was calculated as the number of intraepidermal nerve fibers (Figure 3A, yellow arrowheads) divided by the length of epidermal basement membrane (Figure 3A, white dashed line). Paclitaxel decreased the density of intraepidermal nerve fibers at 4 weeks, but not at 1 week, after starting paclitaxel (Figure 3B,C). Daily oral administration of 50 mg/kg GSSSG prevented paclitaxel-induced loss of intraepidermal nerve fibers at 4 weeks after starting paclitaxel (*p* = 0.0024).

### 3.3. GSSSG Prevented the Paclitaxel-Induced Loss of Unmyelinated Axons in the Sciatic Nerve

To evaluate the impact of paclitaxel on myelinated and unmyelinated axons, we counted the number of axons in the sciatic nerve at 4 weeks after starting paclitaxel. The number of myelinated axons was similar between control, paclitaxel, and paclitaxel with GSSSG groups (Figure 4A,B). The thickness of myelin, calculated using the G-ratio (a ratio of the inner and outer radius), was not affected by paclitaxel either without or with GSSSG (Figure 4C,D). We also examined transmission electron microscopy images (low magnification, ×2200) to determine the number of unmyelinated axons in the sciatic nerve. Unmyelinated (Figure 5A, arrows) and myelinated axons (Figure 5A, arrowheads) were counted and divided by a total number of axons. Paclitaxel tended to decrease unmyelinated axons compared to control mice (*p* = 0.0695) and showed disruption of architecture in the nerve fascicle (Figure 5A, double arrow), indicating axonal degeneration. Compared to mice that received paclitaxel alone, mice that received paclitaxel and GSSSG had a larger ratio of unmyelinated axons in the sciatic nerve (Figure 5A,B, *p* = 0.0153). These results suggest that GSSSG prevents axonal loss of unmyelinated neurons after paclitaxel treatment.

### 3.4. GSSSG Prevented Mitochondrial Swelling in Unmyelinated Sciatic Nerve Axons

To explore the mechanisms by which paclitaxel induces degeneration of unmyelinated axons, we examined mitochondrial morphology in unmyelinated sciatic nerve axons 4 weeks after starting paclitaxel. Mitochondria appeared to be larger and swollen in the unmyelinated axons of paclitaxel-treated mice (Figure 6A, arrowheads) compared to control mice and mice treated with paclitaxel and GSSSG. The cross-sectional area of mitochondria in unmyelinated axons in paclitaxel-treated mice was significantly larger than those in control mice (Figure 6B, *p* < 0.0001) and mice treated with paclitaxel and GSSSG (Figure 6B, *p* = 0.001). These results suggest that the beneficial effects of GSSSG on paclitaxel-induced axonal degeneration are mediated by protection of mitochondrial integrity.

### 3.5. ^34^S-Labeled GSSSG Was Detected in DRG, Spinal Cord, Brain, and Liver after Oral Administration

To study the pharmacokinetics of orally administrated GSSSG, we used ^34^S-labelled GSSSG to determine the distribution of GSSSG and its metabolites in central and peripheral nervous systems. Lumber DRG, lumber spinal cord, brain, liver, and plasma were harvested 2 h after oral administration of ^34^S-labelled GSSSG at 50 mg/kg, and the levels of ^34^S-labelled GSSSG in each tissue were determined by liquid chromatography-tandem mass spectrometry (LC-MS/MS). While endogenous GSSSG was not detected, the average concentration of administrated ^34^S-labelled GSSSG was 415, 518, 142, and 158 pmol/mg protein in lumber DRG, lumber spinal cord, brain, and liver, respectively (Figure 7A). The plasma level of GSSSG was 58 pmol/mL. We also determined the ratio of exogenous (containing ^34^S) to endogenous (containing ^32^S) reactive sulfur species: GSSH, CysSSH, and CysSSSCys, in these 4 tissues (Figure 7B–D, respectively). The concentration of ^34^S-labelled GSSH, CysSSH, and CysSSSCys was more than 10-fold higher than the endogenous levels of GSSH, CysSSH, and CysSSSCys in all four tissues. These observations indicate that orally administered GSSSG was absorbed from the gastrointestinal tract, entered the central and peripheral nervous systems and was partially metabolized to other reactive sulfur species.

### 3.6. GSSSG Prevented Paclitaxel-Induced Axonal Degeneration and Fragmentation of Mitochondria in Cultured Primary DRG Neurons

The effects of GSSSG on axonal integrity were examined in cultured murine primary DRG neurons treated with paclitaxel. Neurons that were incubated with paclitaxel (100 nM) for one-hour showed bulbed axonal endings compared to untreated neurons (Figure 8A arrowheads). The number and morphology of neurons (length, and branching of neurites), were assessed using the Sholl analysis, using ImageJ [31]. The Sholl analysis plugin draws equally spaced circles from the soma and counts the number of intersections between neurites and circles to quantify neuronal morphology (Figure 8B). Incubation with paclitaxel for one-hour inhibited the elongation of axons while co-administration of GSSSG restored axonal elongation (Figure 8A,C). The number of neurites greater than 160 μm from the soma in paclitaxel and GSSSG treated cells was significantly larger than the number in neurites treated with paclitaxel alone (*p* < 0.05). Incubation with paclitaxel also decreased the ratio of the total mitochondria length to axonal length and the number of mitochondria in axons, which are signs of increased mitochondrial fragmentation (Figure 9A). GSSSG prevented the paclitaxel-induced decrease in the ratio of mitochondrial to axonal length (Figure 9B, *p* < 0.0001) and the number of mitochondria (Figure 9C, *p* = 0.0017). These results suggest that the beneficial effects of GSSSG on PIPN are mediated by prevention of axonal degeneration via preservation of mitochondrial integrity in peripheral neural axons.

### 3.7. GSSSG Attenuated Paclitaxel-Induced Increase of Superoxide Levels in Primary Cortical Neurons

To consider the possibility that GSSSG protects axons via an antioxidant effect [15], we examined the change in intracellular ROS in primary cortical neurons. After incubating primary cortical neurons with paclitaxel (100 nM) for one-hour, intracellular levels of superoxide were measured using the Dihydroethidium (DHE). The levels of superoxide increased after incubation with paclitaxel (Figure 10, *p* = 0.01, paclitaxel vs. control). Co-incubation with GSSSG (10 μM) prevented the paclitaxel-induced increase in intracellular superoxide (*p* = 0.0114, paclitaxel vs. paclitaxel + GSSSG). These results suggest that GSSSG attenuates ROS production induced by paclitaxel.

### 3.8. GSSSG Upregulated Antioxidant Signaling in DRG

To further characterize the effects of GSSSG on PIPN, we used real-time qPCR to measure the levels of mRNA encoding antioxidant proteins in lumber DRG, 2 h after a single paclitaxel injection (16 mg/kg) with or without 50 mg/kg GSSSG. Treatment with either paclitaxel or paclitaxel and GSSSG increased Nrf2 mRNA levels (Figure 11A, *p* = 0.0008 control vs. paclitaxel, *p* < 0.0001 control vs. paclitaxel and GSSSG). The levels of mRNAs encoding enzymes that are downstream of Nrf2, including Heme Oxygenase 1 (HO1, Figure 11B), NAD(P)H Quinone Dehydrogenase 1 (NQO1, Figure 11C) and Glutamate-Cysteine Ligase Catalytic Subunit (GCLC, Figure 11D), were increased only in mice that received paclitaxel and GSSSG (*p* = 0.0127, 0.0024 and 0.0038, respectively, control vs. paclitaxel and GSSSG). These results suggest that GSSSG enhances Nrf2-dependent antioxidant signaling in DRG after paclitaxel treatment.

### 3.9. GSSSG Did Not Affect the Anti-Tumor Effects of Paclitaxel in a Human Breast Cancer Cell Line

Because several studies showed that enhanced antioxidant effects contribute to the resistance of cancer cells to chemotherapy [32,33], we assessed whether GSSSG affects the anti-tumor effect of paclitaxel. We examined the in vitro effect of paclitaxel on MDA-MB-231 human breast cancer cell viability. After incubation with paclitaxel for 24 h, the viability of MDA-MB-231 cells was assessed using the LDH Cytotoxicity Detection assay (Figure 12A). The IC_50_ of paclitaxel for MDA-MB-231 cells was 1.66 μM. Based on these results, we applied paclitaxel (2 μM) and GSSSG (10 μM) to MDA-MB-231 cells for 24 h to determine whether co-administration of GSSSG alters the cytotoxic effects of paclitaxel on MDA-MB-231 cells. After 24 h of incubation with paclitaxel alone or paclitaxel with GSSSG, the percentage of viable MDA-MB-231 cells (compared to untreated cells) was 44.9% and 48.3%, respectively, (Figure 12B). The statistical test did not show a difference more than defined 10% margin (mean difference: 0.034, 90% confidence interval: −0.280 to 0.357). These results demonstrate that GSSSG does not inhibit that ability of paclitaxel to kill MDA-MB-231 cells.

## 4. Discussion

The current study revealed that GSSSG, a polysulfide, has the potential to mitigate paclitaxel-induced peripheral neuropathy by protecting mitochondria in the axons of peripheral neurons. This conclusion is based on the following results: (1) daily oral administration of GSSSG attenuated mechanical allodynia in the hind paw of mice treated with paclitaxel; (2) orally administered GSSSG was absorbed and increased reactive sulfur levels in lumbar DRG and spinal cord (both regions which contain the primary sensory neurons that innervate the hind paw); (3) GSSSG inhibited axonal degeneration and prevented mitochondrial swelling in paclitaxel treated mice treated and maintained the number of axonal mitochondria in paclitaxel-exposed cultured primary DRG neurons; and (4) GSSSG attenuated the increase in superoxide levels in primary cortical neurons incubated with paclitaxel. Taken together, these results suggest GSSSG ameliorates PIPN. The beneficial effects of GSSSG were associated with protection of mitochondrial integrity in peripheral nervous system axons.

In the current study, we observed that oral administration of GSSSG at 50 mg/kg/day for 4 weeks ameliorated paclitaxel-induced mechanical allodynia and ^34^S-labeled GSSSG was detected in lumber DRG and lumber spinal cord 2 h after single oral administration. The results show that GSSSG is well absorbed from the gastrointestinal tract and readily taken up by central and peripheral nervous tissues [34]. We also observed that the concentrations of related polysulfides and persulfides containing ^34^S were more than 10-fold greater than those of respective endogenous polysulfides and persulfides 2 h after ^34^S-labeled GSSSG administration. These observations suggest that GSSSG is in dynamic equilibrium with other reactive sulfur species. To the best of our knowledge, this is the first study demonstrating an association between neuroprotective effects and increased levels of polysulfides in the peripheral nervous system after systemic administration of a polysulfide donor.

Impairment of mitochondria and increased oxidative stress play crucial roles in PIPN [35]. Mitochondria are a main source of cellular ROS but are also important targets of ROS. Normal mitochondria release low levels of ROS, as small amounts of electrons leak from complex I, III, and IV of the electron transport chain (ETC) and bind with molecular oxygen to produce superoxide. When electron transport is impaired, more electrons leak from ETC complexes to produce more ROS. Increased ROS production by dysfunctional mitochondria further impairs mitochondrial function, eventually leading to collapse of the mitochondrial membrane potential. Defective mitochondria are degraded and replaced by quality control mechanisms, such as mitophagy. Because neurons predominantly rely on oxidative phosphorylation in mitochondria to produce ATP, mitochondrial dysfunction causes degeneration of neuronal axons [36].

Reactive sulfur species are thought to protect mitochondria through several mechanisms: (1) supporting bioenergetics [37]; (2) scavenging ROS [15]; (3) activating superoxide dismutase (SOD) [38]; (4) preventing fission of mitochondria by inhibiting Drp1 activity [37]; and (5) upregulating the Nrf2/Keap1 pathway by persulfidation of Keap1 [39]. In the current study, administration of GSSSG attenuated paclitaxel-induced ROS production and prevented swelling and loss of mitochondria in peripheral neural axons. These observations suggest that GSSSG has anti-oxidative effects and that GSSSG may support bioenergetics. Our results in primary DRG neurons suggest that GSSSG prevents not only mitochondria fragmentation (or fission) induced by paclitaxel, but also loss of mitochondria in neural axons. As reported previously, redox balance and glutathione oxidation in neural axons partly regulate mitochondrial transportation from cell body to axon [40], and loss of axonal mitochondria results in axonal degeneration [41] and neuropathy [42]. Our observations support that GSSSG attenuates PIPN via protecting mitochondria by regulating redox balance in peripheral neurons. Interestingly, paclitaxel, without or with GSSSG, upregulated Nrf2 a master regulator of cellular homeostasis, redox balance, and inflammation [43], suggesting that paclitaxel triggers an antioxidant defense mechanism. However, we observed that GSSSG with paclitaxel, but not paclitaxel alone, upregulated expression of HO1, NQO1 and GCLC, which are the downstream genes of Nrf2. One possible reason why GSSSG upregulates genes that are downstream of Nrf2 is that GSSSG induces persulfidation of Keap1, a Nrf2 binding protein. Persulfidation of Keap1 promotes activation of Nrf2, thereby upregulating the downstream genes and exerting an antioxidative effect [39]. In addition to the Nrf2 pathway, GSSSG may exert protective effects on peripheral neurons via multiple antioxidant mechanisms.

We recognize several limitations in our current study. First, there is a discrepancy that GSSSG mitigates only mechanical allodynia (a sign of Aβ fiber impairment) but not thermal hyperalgesia (a sign of altered Aδ and C fiber function) whereas it protects unmyelinated fibers (C fibers). This discrepancy was also reported in previous studies [44,45]. While several mechanisms have been proposed to explain the discrepancy between mechanical allodynia and thermal hyperalgesia, one conceivable mechanism is phenotypic switch in which, following injury (in this case chemotherapeutic-induced), Aβ function is altered leading to mechanical allodynia via central secretion of pro-nociceptive compounds [46]. Second, in this study, we used LC-MS/MS to demonstrate that GSSSG is absorbed and distributed in the DRG. However, we did not measure levels of reactive sulfur species, GSH, and cysteine after administration of paclitaxel and/or GSSSG. Dynamic changes in the levels of reactive sulfur species and intracellular thiols in peripheral neurons under paclitaxel-induced oxidative stress remain to be determined. Third, all mice in the current study were male. Although previous studies showed that mechanical allodynia induced by paclitaxel was not affected by gender [47], this was not confirmed in the present study. Lastly, we used qPCR to measure the levels of mRNA encoding proteins involved in the anti-oxidative response. The small size of murine lumbar DRG made it difficult to measure the level of proteins in this pathway.

## 5. Conclusions

In conclusion, the current study showed that oral administration of GSSSG ameliorated paclitaxel-induced mechanical allodynia in a mouse model of PIPN. After oral administration, GSSSG reached DRG and prevented impairment of mitochondria and axonal degeneration in peripheral neurons. We also found that GSSSG upregulated Nrf2-dependent antioxidant signaling pathway in vivo and diminished ROS levels in vitro. Because impairment of mitochondria is a common feature of diverse forms of peripheral neuropathy [42], these results imply potential therapeutic effects of GSSSG, not only for CIPN, but also for other forms of peripheral neuropathy caused by mitochondrial dysfunction, such as diabetic neuropathy.

## Figures and Tables

**Figure 1 antioxidants-11-02122-f001:**
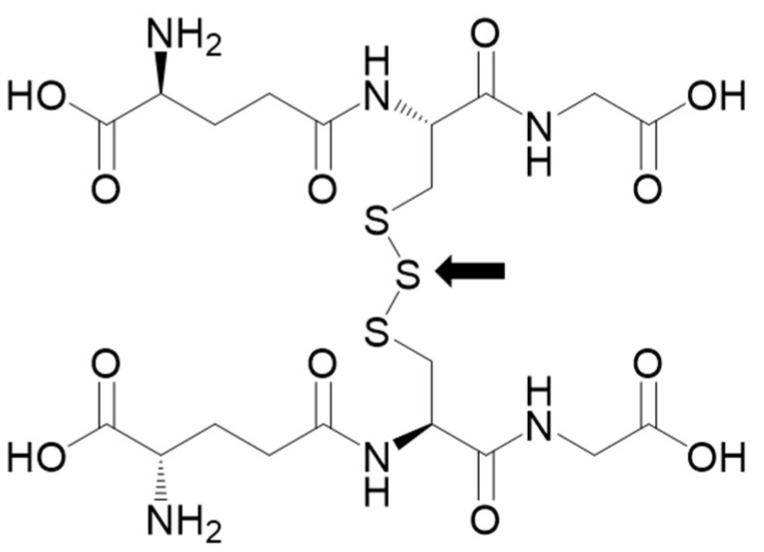
Structural formula of glutathione trisulfide (GSSSG). Purified GSSSG is white, odorless solid powder at 20 °C. Molecular weight is 644.7. It contains one sulfane sulfur (arrow).

**Figure 2 antioxidants-11-02122-f002:**
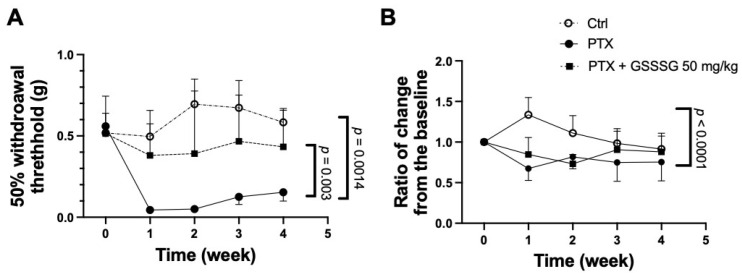
GSSSG prevented mechanical allodynia evoked by paclitaxel-induced peripheral neuropathy. (**A**) Mechanical withdrawal threshold by the von Frey test for 4 weeks after paclitaxel treatment with or without co-treatment of GSSSG. 50 mg/kg of GSSSG prevented mechanical allodynia over the experimental period. (**B**) Relative change of response time compared to the baseline in hot plate test for 4 weeks. PTX induced thermal hyperalgesia. Data are analyzed by mixed effect model and adjusted by Bonferroni correction. Data are shown as the mean ± SD, *n* = 6 mice per group. PTX, paclitaxel.

**Figure 3 antioxidants-11-02122-f003:**
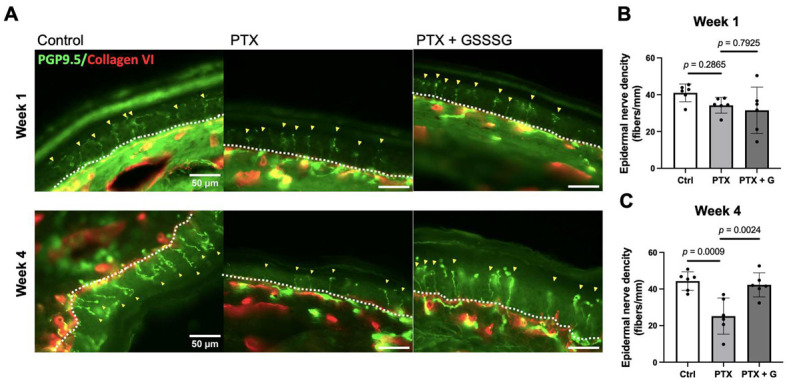
(**A**) Representative immunofluorescence images of intraepidermal nerve fibers (stained by PGP9.5, shown by yellow arrowheads) and basement membrane (stained by Collagen IV, white dashed lines) in hind paw at 1 week and 4 weeks after paclitaxel treatment. Quantification of unmyelinated fiber density calculated by dividing number of intraepidermal nerve fibers by basement membrane at 1 week (**B**) and at 4 weeks (**C**). Data were analyzed by one-way ANOVA with Dunnett’s multiple comparisons test. Individual data (black dots) are shown with mean ± SD, *n* = 6 mice per group. PTX, paclitaxel; G, GSSSG.

**Figure 4 antioxidants-11-02122-f004:**
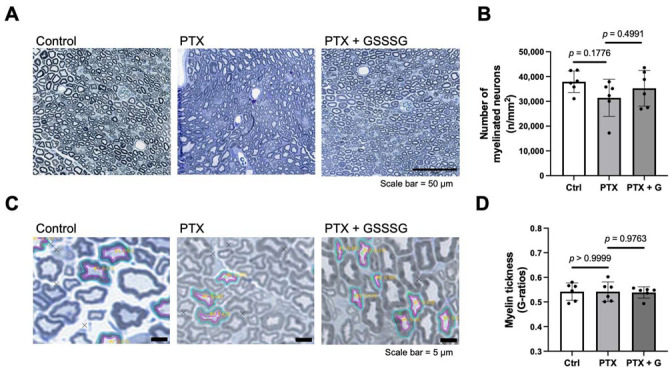
Neither PTX nor GSSSG changed number of myelinated axons. (**A**) Representative microscopic images of sciatic nerve at 4 weeks after PTX treatment stained with toluidine blue. (**B**) Quantification of total myelinated neurons axons in sciatic nerve. (**C**) Representative high magnification images of myelinated neurons axons to measure thickness of myelin. The thickness of myelin was calculated by GRatio software, an ImageJ plugin in ten neurons per image randomly selected by cross marks generated by GRatio. (**D**) Quantification of G-ratios. Neither PTX nor GSSSG did not change thickness of myelin. Data were analyzed by one-way ANOVA with Dunnett’s multiple comparisons test. Individual data (black dots) are shown with mean ± SD, *n* = 6 mice per group. PTX, paclitaxel; G, GSSSG.

**Figure 5 antioxidants-11-02122-f005:**
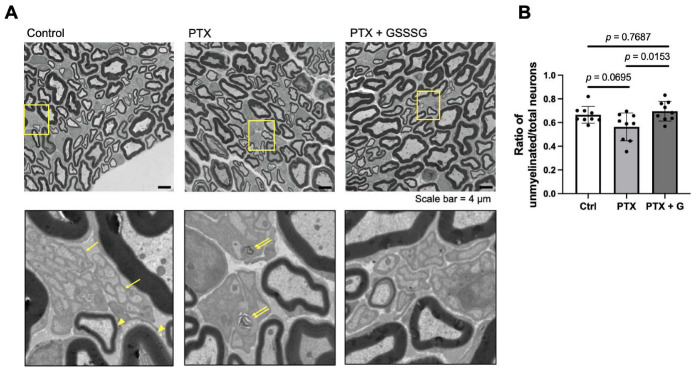
GSSSG prevented loss of unmyelinated axons in sciatic nerve. (**A**) Representative transmission electron microscopy images of sciatic nerve at 4 weeks. Images in the second row are magnified images of the area indicated by yellow frames in the first row image of respective column. Number of unmyelinated axons (yellow arrows) and myelinated axons (yellow arrowheads) were counted. (**B**) Ratio of the number of unmyelinated axons of the total number of axons in sciatic nerve at 4 weeks. Data were analyzed by one-way ANOVA with Tukey’s multiple comparisons test. Individual data (black dots) are shown with mean ± SD, *n* = 3 mice per group. PTX, paclitaxel. G, GSSSG.

**Figure 6 antioxidants-11-02122-f006:**
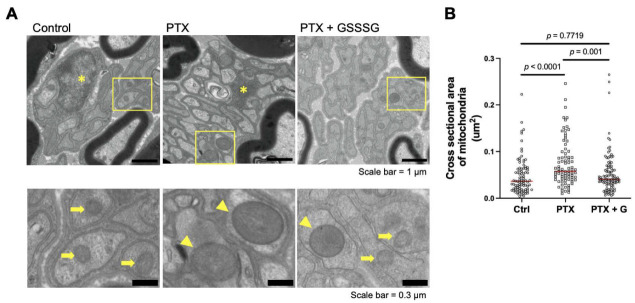
GSSSG prevented degeneration of mitochondria. (**A**) Representative transmission electron microscopy images of sciatic nerve at 4 weeks after paclitaxel treatment. Images in the second row are magnified images of the area indicated by yellow frames in the first row image of respective column. Asterisks show Schwann cell, arrows indicate mitochondria and arrowheads indicate swollen mitochondria. (**B**) Quantification of mitochondria area in unmyelinated neuron. The median area of mitochondria in control, paclitaxel, and paclitaxel with GSSSG were 0.036 μm^2^, 0.057 μm^2^, and 0.041 μm^2^, respectively. Red bars indicate median values of each group. Data were analyzed by Kruskal–Wallis test with Dunn’s multiple comparisons test. Four high (×11,000) magnification images from each mouse were examined. *n* = 3 mice per group. PTX, paclitaxel; G, GSSSG.

**Figure 7 antioxidants-11-02122-f007:**
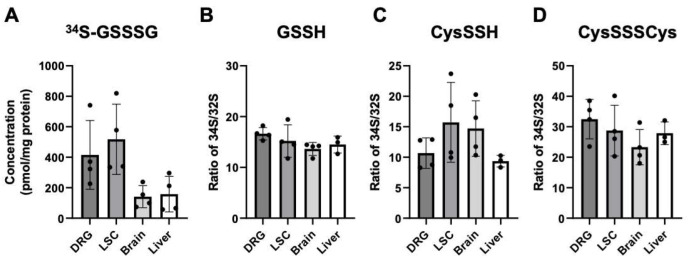
^34^S-labeled GSSSG and related polysulfides were detected in peripheral tissues 2 h after oral administration of ^34^S-labeled GSSSG. Quantification of GSSSG and reactive sulfur species by LC-MS/MS. (**A**) ^34^S-labeled GSSSG concentration in 4 tissues that were obtained 2 h after oral administration. Relative ratio of ^34^S-laveled reactive sulfur species to endogenous reactive sulfur species (^32^S-) were calculated. The ratios of ^34^S-laveled to endogenous glutathione persulfide (GSSH) (**B**), cysteine persulfide (CysSSH) (**C**), cysteine trisulfide (CysSSSCys) (**D**) in lumber DRG, lumber spinal cord, brain, and liver were reported. Individual data (black dots) are shown with mean ± SD, *n* = 4 per group; DRG, dorsal root ganglion; LSC, lumber spinal cord.

**Figure 8 antioxidants-11-02122-f008:**
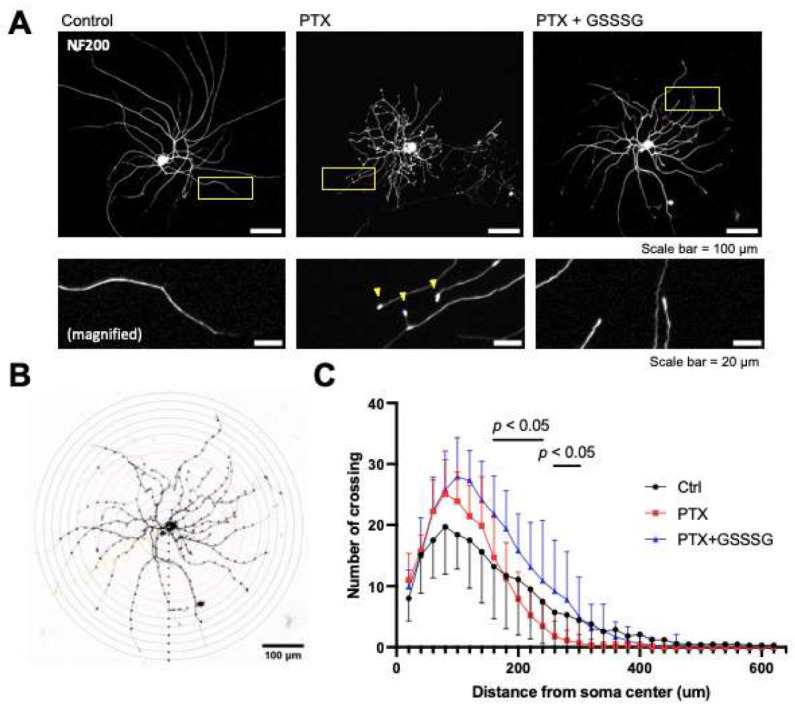
GSSSG prevented axon degeneration in primary DRG neurons. (**A**) Representative immunofluorescence images of primary DRG neurons stained by NF200. Neurons were cultured for 24 h and incubated by paclitaxel with or without GSSSG for one hour. (**B**) Sholl circles of cultured neuron. The interval between Sholl circles is 10 μm. (**C**) Analysis of neuronal intersections with Sholl circles. Data were analyzed by repeated ANOVA with Dunnett’s multiple comparisons test. *p* values are shown to indicate the comparison between paclitaxel and paclitaxel + GSSSG. Data are shown as the mean ± SD, *n* = 3 mice per group, 9–10 neurons per group. PTX, paclitaxel.

**Figure 9 antioxidants-11-02122-f009:**
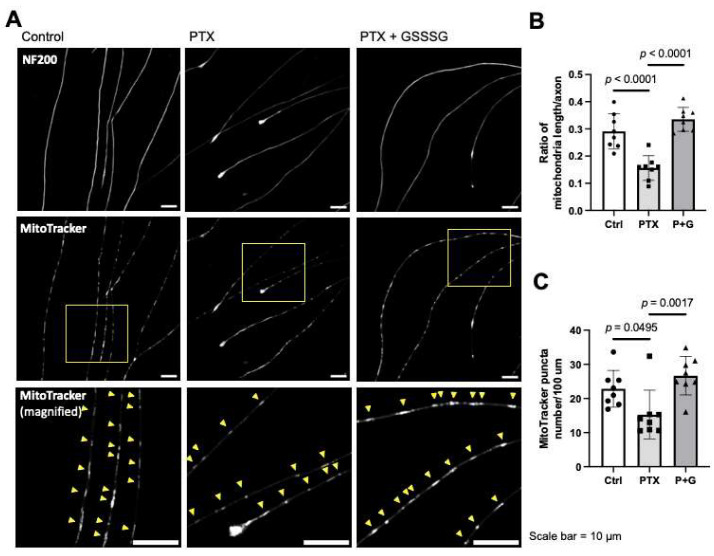
GSSSG protected mitochondria in axon of primary DRG neurons. (**A**) Representative images of neural axons stained by NF200 with axonal mitochondria stained by MitoTracker. Images in the second row are magnified images of the area indicated by yellow frames in the first row image of respective column. The number and length of mitochondria (yellow arrowheads) in axon was measured manually. (**B**) Ratio of the total mitochondria length to the total axonal length and (**C**) The number of MitoTracker puncta per 100 μm of axon. Data were analyzed by one-way ANOVA with Dunnett’s multiple comparison. Individual data (black dots) are shown with mean ± SD, *n* = 3 mice per group, 8 images per group, and 450–790 mitochondria per group. PTX, paclitaxel; G, GSSSG.

**Figure 10 antioxidants-11-02122-f010:**
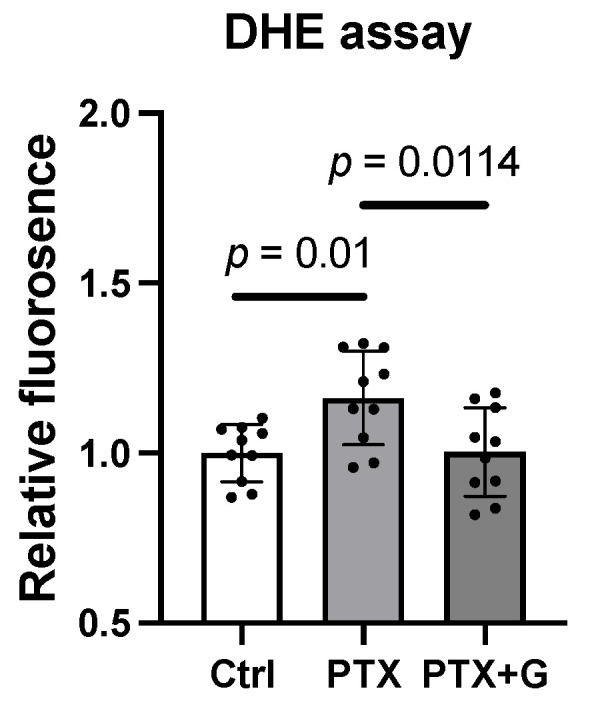
GSSSG treatment blocked increase of superoxide. Relative fluorescence change of Dihydroethidium (DHE) 30 min incubation of GSSSG and 60 min after paclitaxel exposure. Data were analyzed by one-way ANOVA with Dunnett’s multiple comparisons test. Individual data (black dots) are shown with mean ± SD, *n* = 8 per group. PTX, paclitaxel; G, GSSSG.

**Figure 11 antioxidants-11-02122-f011:**
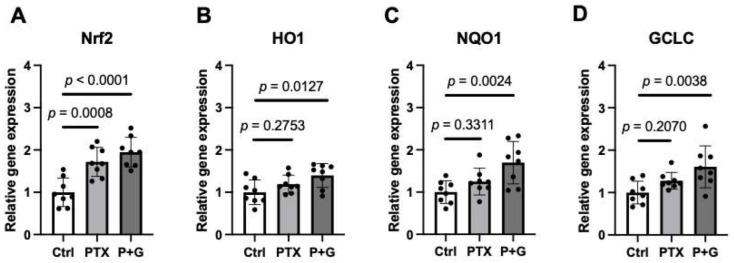
GSSSG upregulated antioxidant signaling in DRG. Relative gene expression of Nrf2 (**A**), HO1 (**B**), NQO1 (**C**), and GCLC (**D**) in DRG tissue at 2 h after single 16 mg/kg paclitaxel with or without 50 mg/kg GSSSG treatment. Data were analyzed by one-way ANOVA with Dunnett’s multiple comparisons test. Data are shown as the mean ± SD, *n* = 8 mice per group. PTX, paclitaxel; G, GSSSG; Nrf2, Nuclear factor-erythroid factor 2-related factor 2; HO1, Heme Oxygenase 1; NQO1, NAD(P)H Quinone Dehydrogenase 1; GCLC, Glutamate-Cysteine Ligase Catalytic Subunit.

**Figure 12 antioxidants-11-02122-f012:**
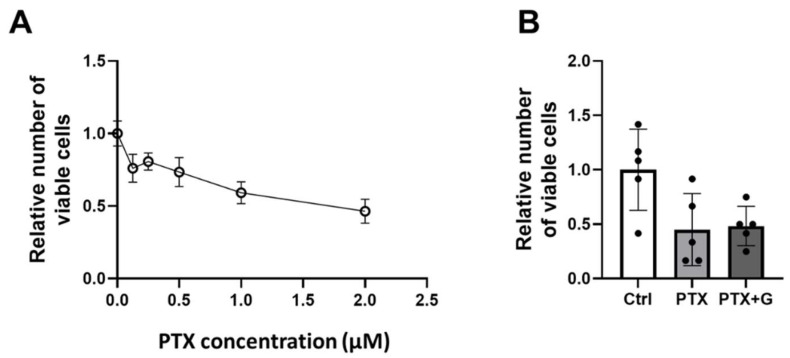
Co-administration of GSSSG did not inhibit anti-tumor effect of paclitaxel. (**A**) Cell viability of human breast cancer cell line, MDA-MB-231 measured by LDH assay at 24 h after PTX with/without 10 μM of GSSSG. (**B**) Relative viable numbers of MDA-MB-231 cells 24 h after 2 μM of PTX with/without 10 μM of GSSSG. Data were analyzed by two one-sided *t*-tests. The margin of equivalence was defined as 10% difference in cell count. Data are shown as the mean ± SD, *n* = 5 per group; LDH, lactate dehydrogenase, PTX, paclitaxel; G, GSSSG.

## Data Availability

The data presented in this study are available on request from the corresponding author.

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
