# Peer review of "Oral Administration of Glutathione Trisulfide Increases Reactive Sulfur Levels in Dorsal Root Ganglion and Ameliorates Paclitaxel-Induced Peripheral Neuropathy in Mice"

_antioxidants, 2022, doi:10.3390/antiox11112122_

Round 1
Reviewer 1 Report
In this manuscript, the authors examined neuroprotective effect of glutathione trisulfide (GSSSG) in mouse model of paclitaxel-induced peripheral neuropathy (PIPN). Oral administration of GSSSG ameliorated mechanical allodynia but not thermal hyperalgesia in mouse model. The authors also found that GSSSG prevented paclitaxel-induced loss of unmyelinated axons and degradation of mitochondria in the sciatic nerve. They also examined distribution of GSSSG and its metabolites by using 34S-labelled GSSSG, and found that orally administered 34S-labeled GSSSG was absorbed and uptaken into central and peripheral nervous systems and metabolized to other reactive sulfur species (GSSH, CysSSH, CysSSSCys). Furthermore, they showed that GSSSG upregulated antioxidant proteins and mitigated paclitaxel-induced superoxide production. From these evidences, they concluded that GSSSG mitigated PIPN by preventing axonal degeneration and mitochondria damage in peripheral sensory nerves. This is interesting manuscript and might be contributed to the possible therapy reactive sulfur donors. However, there are some concerns and points of doubt as listed below.
1. The authors examined quantitative analysis of reactive sulfur species in tissues by using LC-MS/MS. However, they showed only quantification of 34S labeled GSSSG after oral administration of GSSSG in mouse tissues. Do reactive sulfur levels degrease in paclitaxel treatment, and recover by treatment with GSSSG? Also, intracellular levels of GSH and Cysteine might affect by administration of GSSSG.
2. Details information of 34S labeled GSSSG is needed. Which sulfur in GSSSG is labeled 34S?
Author Response
Comment from Reviewer 1
- The authors examined quantitative analysis of reactive sulfur species in tissues by using LC-MS/MS. However, they showed only quantification of 34S labeled GSSSG after oral administration of GSSSG in mouse tissues. Do reactive sulfur levels decrease in paclitaxel treatment, and recover by treatment with GSSSG? Also, intracellular levels of GSH and Cysteine might affect by administration of GSSSG.
The purpose of this experiments was to examine whether oral administration of GSSSG is absorbed and distributed in dorsal root ganglion. Although dynamic changes of levels of reactive sulfur species, GSH, and cysteine in response to paclitaxel and GSSSG are of interest to us, the miniscule amount of mouse DRG tissue limits what we could measure. Nonetheless, we added a paragraph in limitation section to acknowledge this point.
Page 15, line 670
We recognized several limitations…
Second, in this study, we used LC-MS/MS to demonstrate that GSSSG is absorbed and distributed in the DRG. However, we did not measure levels of reactive sulfur species, GSH, and cysteine after administration of paclitaxel and/or GSSSG. Dynamic changes in the levels of reactive sulfur species and intracellular thiols in peripheral neurons under paclitaxel-induced oxidative stress remain to be determined. Third, ...
- Details information of 34S labeled GSSSG is needed. Which sulfur in GSSSG is labeled 34S?
Thank you for your pointing this out. We have explained detail information of 34S-laveled GSSSG in the method part.
Page 5, line 230-
We used liquid chromatography with tandem mass spectrometry (LC-MS/MS) to examine whether administrated GSSSG via oral gavage reaches the peripheral nervous system. Fifty mg/kg of 34S-labeled GSSSG (in which the middle sulfur, that is the sulfane sulfur, was replaced with 34S; G-32S-34S-32S-G) was orally administered to mice.
Reviewer 2 Report
Review for Manuscript Antioxidants-1960489, “Oral administration of glutathione trisulfide increases reactive sulfur levels in dorsal root ganglion and ameliorates paclitaxel induced peripheral neuropathy in mice”.
In the present study, the authors examined neuroprotective effects of glutathione trisulfide (GSSSG) in a mouse model of PIPN
Summary of results and significance:
This study represents an impressive amount of work, is quite thorough, and the authors’ claims are well-substantiated based on multiple types of supporting evidence. The authors showed that oral administration of glutathione trisulfide (GSSSG) mitigated paclitaxel-induced peripheral neuropathy (PIPN) by preventing axonal degeneration and mitochondria damage in peripheral sensory nerves. The data within the paper are well-presented, clear, and thoroughly analyzed. However, there are some concerns with the lack of key experiments that decrease its scientific impact.
Major critiques:
1. The authors observed that GSSSG with paclitaxel, not paclitaxel alone, upregulated expression of HO1, NQO1 and GCLC which are the downstream genes of Nrf2. To substitute their claim, the authors should investigate further how GSSSG regulate the downstream targets of Nrf2 to have a better understanding for the readers the antioxidant mechanism of protection of GSSSG.
2. According to the authors, the study was designed to examine the effects of a systemically administered stable formulation of glutathione trisulfide (GSSSG), an endogenous polysulfide, in a mouse model of PIPN. GSSSG contains sulfane sulfur in the structure and it is in an equilibrium between GSSH in the following equation: GSSSG + GSH ↔ GSSH + GSSG.
Under oxidative stress, where GSH is oxidized to GSSG, how will GSSSG interact with GSH? What will happen when GSSG to GSH ratio is increased? The authors should discuss how will the equilibrium be maintained by GSSSG under oxidative stress?
Author Response
Comment from Reviewer 2
Thank you for your insightful comments. You checked “English language and style are fine/minor spell check required”. To improve English, we asked one of our colleagues, Dr. Donald B Bloch, to review and edit the manuscript. We have added him as a co-author and submitted Authorship change form with this reply.
- The authors observed that GSSSG with paclitaxel, not paclitaxel alone, upregulated expression of HO1, NQO1 and GCLC which are the downstream genes of Nrf2. To substitute their claim, the authors should investigate further how GSSSG regulate the downstream targets of Nrf2 to have a better understanding for the readers the antioxidant mechanism of protection of GSSSG.
Thank you for your comments. Aside from upregulating Nrf2, we are considering the possibility that GSSSG upregulated downstream genes of Nrf2 by affecting Keap1 according to the previous studies. We added a few sentences to explain this in the discussion.
Page 15, line 656
Interestingly, paclitaxel, without or with GSSSG, upregulated Nrf2 a master regulator of cellular homeostasis, redox balance, and inflammation [43], suggesting that paclitaxel triggers an antioxidant defense mechanism. However, we observed that GSSSG with paclitaxel, but not paclitaxel alone, upregulated expression of HO1, NQO1 and GCLC, which are the downstream genes of Nrf2. One possible reason why GSSSG upregulates genes that are downstream of Nrf2 is that GSSSG induces persulfidation of Keap1, a Nrf2 binding protein. Persulfidation of Keap1 promotes activation of Nrf2, thereby upregulating the downstream genes and exerting an antioxidative effect [39]. In addition to the Nrf2 pathway, GSSSG may exert protective effects on peripheral neurons via multiple antioxidant mechanisms.
- According to the authors, the study was designed to examine the effects of a systemically administered stable formulation of glutathione trisulfide (GSSSG), an endogenous polysulfide, in a mouse model of PIPN. GSSSG contains sulfane sulfur in the structure and it is in an equilibrium between GSSH in the following equation: GSSSG + GSH ↔ GSSH + GSSG.
Under oxidative stress, where GSH is oxidized to GSSG, how will GSSSG interact with GSH? What will happen when GSSG to GSH ratio is increased? The authors should discuss how will the equilibrium be maintained by GSSSG under oxidative stress?
Thank you for your comment. Because intracellular levels of GSH is in mM while GSSSG is nM or less, it is likely that there are sufficient GSH exist enough to interact with GSSSG even under oxidative stress. Nonetheless, we did not measure GSH levels after administration of paclitaxel and or GSSSG and this deficit is acknowledged in the limitation section of the Discussion. Please see our response to reviewer 1 comment #1.
Round 2
Reviewer 2 Report
The authors have addressed to the reviewer's comments. The manuscript is suitable for acceptance in its present form.